

# Establishment and validation of an individualized macrophage-related gene signature to predict overall survival in patients with triple negative breast cancer

Peng Su[1,*], Ziqi Peng[1,*], Boyang Xu[2], Bowen Yang[3] and Feng Jin[1]

[1] Department of Breast Surgery, The First Affiliated Hospital of China Medical University, Shenyang, Liaoning, China
[2] Gastroenterology Department, The First Affiliated Hospital of China Medical University, Shenyang, Liaoning, China
[3] Department of Medical Oncology, The First Affiliated Hospital of China Medical University, Shenyang, Liaoning, China
[*] These authors contributed equally to this work.

Corresponding authors
Bowen Yang,
2017110177@cmu.edu.cn
Feng Jin, jinfeng@cmu.edu.cn

## ABSTRACT

**Background.** Recently, researchers have classified highly heterogeneous triple negative breast cancer (TNBC) into different subtypes from different perspectives and investigated the characteristics of different subtypes to pursue individualized treatment. With the increase of immunotherapy and its preliminary application in TNBC treatment, the value of immune-related strategies in the treatment of TNBC has been initially reflected. Based thereon, this study plans to classify and further explore TNBC from the perspective of immune cell infiltration.

**Method.** The fractions of immune cells of TNBC patients were assessed by six immune component analysis methods in The Cancer Genome Atlas (TCGA) database. Hub genes significantly related to poor prognosis were verified by weighted gene co-expression network analysis (WGCNA) analysis, Lasso analysis, and univariate KM analysis. Two cohorts of TNBC patients with complete prognosis information were collected for validation analysis. Finally, the Genomics of Drug Sensitivity in Cancer (GDSC) database was adopted to ascertain the sensitivity differences of different populations based on hub-gene grouping to different chemotherapy drugs.

**Results.** Five hub genes (CD79A, CXCL13, IGLL5, LHFPL2, and PLEKHF1) of the key co-expression gene module could divide TNBC patients into two groups (Cluster A and Cluster B) based on consistency cluster analysis. The patients with Cluster A were responsible for significantly worse prognosis than the patients with Cluster B ($P = 0.023$). In addition, another classification method, PCoA, and two other datasets (GSE103091 and GSE76124), were used to obtain consistent results with previous findings, which verified the stability of the classification method and dataset in this study. The grouping criteria based on the previous results were developed and the accuracy of the cut-off values was validated. A prognosis model of TNBC patients was then constructed based on the grouping results of five hub genes and N staging as prognostic factors. The results of ROC and decision curve analyses showed that this model had high prediction accuracy and patients could benefit therefrom. Finally,

GDSC database analysis proved that patients in Cluster A were more sensitive to Vinorelbine. Separate analysis of the sensitivity of patients in Cluster A to Gemcitabine and Vinorelbine showed that the patients in Cluster A exhibited higher sensitivity to Vinorelbine. We hypothesized that these five genes were related to gemcitabine resistance and they could serve as biomarkers for clinical drug decision-making after anthracene resistance and taxane resistance in patients with advanced TNBC.
**Conclusion**. This study found five hub prognostic genes associated with macrophages, and a prognostic model was established to predict the survival of TNBC patients. Finally, these five genes were related to gemcitabine resistance in TNBC patients.

## INTRODUCTION

Breast cancer is the most common malignant tumor in women around the world. The latest global cancer data released by the International Agency for Research on Cancer (IARC) of the World Health Organization in 2020 show that the proportion of new cases of breast cancer accounts for 11.7% of all malignant tumors, surpassing lung cancer for the first time and becoming the cancer with the highest incidence in the world. Triple-negative breast cancer (TNBC) accounts for about 15% to 20% of the total number of breast cancers, and it is characterized by strong heterogeneity, high degree of malignancy, strong invasive ability, and short overall survival (OS) time (*Chen, 2019*; *Sharma, 2016*). Endocrine therapy and HER-2 targeted therapy do not work on TNBC due to the deletion of ER, PR, and HER-2 receptors thus leaving chemotherapy as the only option for adjuvant therapy. Sensitivity to chemotherapy population scale is small, about one in three TNBC patients experienced some degree of resistance (*Hutchinson, 2014*; *Szekely, Silber & Pusztai, 2017*).Therefore, the treatment of TNBC has been among the most difficult problems in breast cancer treatment. In recent years, immunotherapy has become the focus of much research into cancer treatment, among which, TNBC was found to be the most immunogenic subtype of breast cancer (*Yu et al., 2017*), and also the subtype that could benefit most from immunotherapy. Based on the results of the Impassion130 clinical trial, the FDA has approved Atirizumab (PD-L1 inhibitor) in combination with albumin paclitaxel for the treatment of advanced TNBC with positive PD-L1 expression (*Schmid et al., 2019*). In addition, Impassion031 (*Harbeck et al., 2020*), Keynote-355 (*Cortes et al., 2020*), and other large clinical studies on TNBC immunotherapy suggest that the survival of patients with TNBC can be improved to a certain extent with the aid of immunotherapy.

Tumor associated macrophages (TAMs) are macrophages infiltrating tumor tissues and are usually the most abundant when infiltrating white blood cells in most tumors; they play an important role in the interaction between tumor cells and the tumor microenvironment (*Cortés et al., 2017*). TAMs were found to participate in the occurrence and development of breast cancer by promoting breast cancer cell proliferation

(*Hao et al., 2012*), promoting breast cancer angiogenesis (*Werno et al., 2010*; *Tsutsui et al., 2005*), mediating immunosuppression (*Vasievich & Huang, 2011*), and inducing epithelial mesenchymal transformation and early metastasis (*Harney et al., 2015*) Furthermore, TAM also plays a role in drug resistance and can activate the specific pathway of tumor stem cells to enhance drug resistance by releasing milk fat globulin (EGF-VIII). MFGE-8 and IL-6 produced by TAM can co-regulate the tumorigenesis and drug resistance of tumor stem cells (*Schwitalla et al., 2013*). Currently, many therapeutic strategies aimed at inhibiting TAM recruitment(*Song, Cui & Yin, 2018*), inhibiting or consuming TAM (*Rumney et al., 2017*) and phenotypic remodeling of TAM (*Lei et al., 2018*) have attracted attention among researchers investigating breast cancer immunotherapy. Over the years, researchers have been committed to dividing highly heterogeneous TNBC into different subtypes from different perspectives and exploring the clinical characteristics of different subtypes to pursue individualized treatment. Different perspectives can also lead to exploration of TNBC in many ways. Based on the aforementioned background, we believe that immune infiltration is a good starting point. From the perspective of immune cell infiltration,a type of tumor-associated immune cell, TAM which is closely related to the development of TNBC, was identified in this study. Furthermore, we hope to classify TNBC by screening hub genes related to macrophages, to predict the survival of patients under different classifications and their sensitivity to chemotherapy drugs, which can provide some reference for clinical individualized medication diagnosis and treatment: this also provides some ideas for future treatment strategies related to macrophages in TNBC. The flow chart through the present study is provided in Fig. 1.

# MATERIALS & METHODS

## Study design and patients

In this study, data of triple-negative breast cancer samples and related clinical pathological information were downloaded from the TCGA database (https://portal.gdc.cancer.gov/). All downloaded sample data meet the following criteria: (a) patients with mRNA expression data and clinical data (b) patients have been diagnosed with TNBC before May 2019 (c) the patient has completed the standardized diagnosis and treatment of breast cancer, including surgery, chemotherapy and radiotherapy (d) it have survival data of the patients with a survival time greater than 10 days. Information from the TCGA database of TNBC patients was used for analysis of immune cell infiltration. In addition, the validation sets GSE103091 and GSE76124 are from the GEO database. Inclusion criteria were as follows: (a) mRNA expression data were available and (b) more than 100 TNBC samples were available with complete patient prognostic information. Bulk data from the GEO database were used to verify the conclusions of the TCGA database analysis.Single-cell correlation analysis data were obtained from public datasets GSE118389 and GSE161529 in GEO database,These two single-cell datasets were used to verify the correlation between the screened hub genes and macrophages.

Step 1. Data preprocessing and macrophages-related hub gene Module

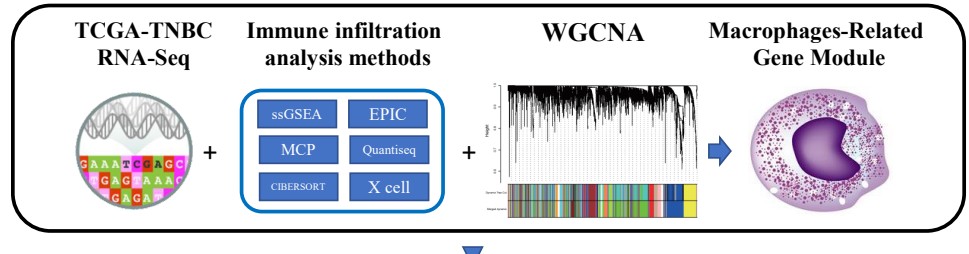

Step 2. Identify 5 hub genes and prognostic analysis

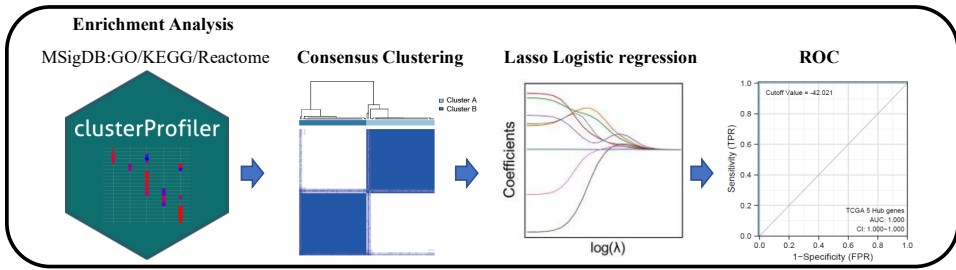

Step 3. Validation analysis

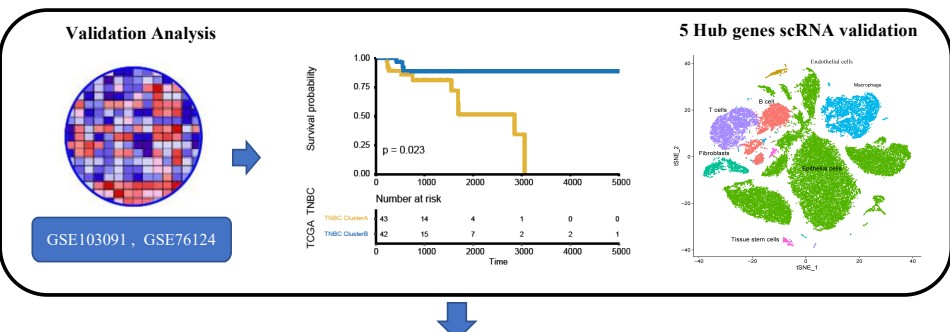

Step 4. Building TNBC prognostic nomogram model and drug sensitivity analysis

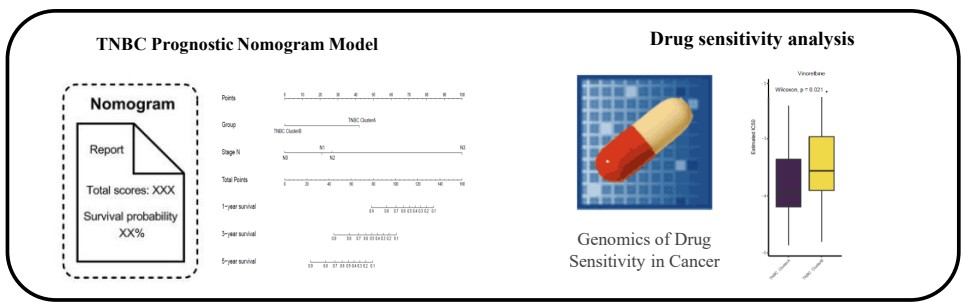

**Figure 1 The flowchart of this study.**

## Immune component analysis of TNBC data

In this study, six immune infiltration analysis methods were used to analyze the immune components of TNBC dataset. A brief introduction will be given below. In SSGSEA algorithm, Rank values of 24 immune cells were calculated based on transcriptome analysis data, and the score of each immune cell in TNBC data of TCGA was obtained through subsequent statistical analysis (*Bindea et al., 2013*). EPIC (http://epic.gfellerlab.org/) can analyze the infiltration ratio of 8 kinds of immune cells, including B cells, macrophages, and NK cells, according to the expression data of TNBC patients in TCGA database. MCP Counter is an R package that quantifies the absolute abundance of 8 immune cells and 2 stromal cells in heterogeneous tissues using transcriptome data; Quantiseq is based on a deconvolution algorithm that uses transcriptome data to predict the composition of 10 types of immune cells in tumor samples. CIBERSORT (https://cibersort.stanford.edu/), which is based on the principle of the linear support vector regression on immune cell subtype of deconvolution of the expression of matrix a tool that can be used to estimate the immune cell infiltration, X cell can integrate the advantages of gene enrichment analysis through d-econvolution to evaluate a variety of immune cells (*Aran, Hu & Butte, 2017*). The aim of this study is to obtain a more accurate and rigorous result by integrating various immune infiltration analysis methods.

## Definition of immune cells related genes in TNBC by WGCNA and enrichment analysis

To define the genes associated with macrophages in breast cancer, we used the Weighted gene co-expression network analysis (WGCNA) First, the first 5,000 genes after the mean absolute deviation (MAD) sequencing and constructed a co-expression network for mRNA expression of these genes using R package- WGCNA. Then, we used the "pickSoftThreshold" function to select the optimal soft threshold parameter to match the gene distribution to a connection-based scale-free network. Then, the adjacencies were transformed into topological overlap matrix (TOM) and the genes with similar expression patterns were divided into the same module. We selected a gene tree with a minimum (genome) of 30 and a module tree with a tangent of 0.25 to obtain different modules, and modules with similar gene distribution were combined into new modules.

After the WGCNA analysis, the correlation analysis was conducted between the scores of various immune cells in the above six methods of immune infiltration analysis and the co-expression modules enriched by WGCNA, hoping to find a gene co-expression module related to one of the immune cells in the six algorithms. Subsequently, Gene Ontology (GO), Kyoto Encyclopedia of Genes and Genomes (KEGG) and Reactome databases were used for enrichment analysis, so as to more accurately judge the biological function and pathway of genes in this module.

## Screening of key genes and construction

Consensus Clustering Analysis is an algorithm that can be used to identify members of clusters in a data set and their number. In this study, Consensus Cluster Plus package was applied to carry out congruent cluster analysis on the genes within the screened
co-expression modules, and Kaplan–Meier survival analysis was applied to determine the correlation between different clusters and the prognosis of TNBC patients. Gene significance (GS) indicates the correlation between individual genes and target immune cell scores. Based on the results of WGCNA analysis, we selected the genes with a threshold of *p* value of GS <0.001 in all the immune infiltration algorithms and *p* value of K-M survival analysis<0.05, that resulted in identification of 12 candidate genes. Subsequently, 1000 Lasso analyses were performed to screen out the most stable genes as hub genes. Consistency cluster analysis was performed on the finally selected hub genes to determine the correlation between different clusters and the survival of TNBC.

## Validation of key genes that have been screened

First, we verify the reliability of the clustering analysis method. PCoA analysis was applied to verify the results of the previous conformance analysis, so as to prove that the hub gene we screened could well divide TNBC patients into two categories. Then, we verified the reliability of the dataset. We selected TNBC datasets GSE103091 and GSE76124 from the GEO database to verify the hub genes-based grouping and prognosis correlation. So far, the stability of the results has been verified in different clustering algorithms and different data sets.

## Development of TNBC patient grouping criteria based on hub gene

Multivariate Cox regression analysis method was used to construct five hub genes related scores of TNBC patients, and the appropriate cut-off value was selected according to the previous PCoA grouping results. TCGA-TNBC were used as the training dataset, and GSE103091 and GSE76124 were used as the validation datasets. The area under The ROC curve (AUC) was used to test the accuracy of the cut-off value. It was used to prove that this value can accurately distinguish patients in all three datasets.

## Correlation verification of HUB gene and macrophages based on single cell sequencing

First, we carried out data quality control and normalization for single-cell dataset, and the process was as follows: (1) Barcode was used to remove the confused cells during the sequencing process. (2) Cells with less than 100 genes identified by sequencing were cleared. (3) calculateQCMetrics function in scater package was used to calculate the proportion of mitochondrial and ribosomal genes expressed in each cell. Remove the low levels cell with mitochondrial gene expression >5% and ribosomal gene expression <10%. (4) After cell quality control, the coefficient matrix was standardized by NormalizeData in the Seurat package. (5) Batch effect between data sets was controlled and eliminated by R package (Harmony).

Then, cell classification and annotation were carried out: 2,000 genes with the largest variation among cells were screened out by the FindVariableFeatures function in Seurat package.ScaleData function was used to linearly standardize data, and RunPCA function was used to perform linear dimensionality reduction analysis. Inflection point values of the decreasing trend of the standard deviation of the principal components in the gravel map were observed as the number of principal components included in subsequent analysis.

Then select the one with large standard deviation (cumulative standard deviation > 70%) of the principal component (PC). The FindNeighbors and FindClusters functions in the Seurat package are used for cell cluster analysis. Then, t-distributed random neighbor embedding (T-SNE) is performed using the RunTSNE function.Finally through the SingleR to annotation of cells, and through the review of previous literature and CellMarker database (http://biocc.hrbmu.edu.cn/CellMarker/) marker genes in various cell types of validate the result of the cell comments.

Finally, the intercellular communication was analyzed: to investigate potential interactions between different cell types in TNBC, cell-to-cell communication analysis was performed using THE R packet (CellChat), a database of literature on mouse and human ligand–receptor interactions that manually reviewed other publicly available signaling pathways, And peer-reviewed literature. Most ligand–receptor interactions in CellChatDB are manually managed on the basis of the KEGG signaling pathway database. In this study, the interaction between macrophages, B cells and T cells was analyzed. The results of the analysis were shown by R packet (igraph).

## Establishment and test of prognosis model of TNBC patients based on hub gene

The relationship between clinical pathological parameters and prognosis of patients with triple-negative breast cancer was analyzed by univariate Cox regression analysis. The parameters with $P < 0.2$ in univariate Cox regression analysis were included in multivariate Cox regression analysis. Finally, the parameters with $P < 0.05$ in multivariate Cox regression analysis were included in the prediction model. The prediction model of prognosis was established by R packet (RMS) and visualized. The primary purpose of this study was to predict the 1-year survival rate, and the secondary purpose was to predict the 3-year and 5-year survival rates.

The area under the ROC curve (AUC) was used to test the differentiation of the prognostic model, which proved that the model could well distinguish patients according to the differences in prognosis over the next few years. The calibration curve is used to test the calibration degree of the model. The calibration degree can indicate the difference between the predicted value and the true value of the model. When there is no statistical difference between the model's calibration degree and the true value ($P > 0.05$), the model is good. Decision Curve Analysis (DCA) is used to evaluate the availability and effectiveness of prediction models. It is equivalent to introducing loss function based on regression prediction analysis. This study was evaluated and visualized using STDCA (https://www.rdocumentation.org) .

## Drug sensitivity analysis based on GDSC database

This study based on GDSC database (https://www.cancerrxgene.org/), the database is stored in the more than two hundred anti-cancer drug sensitivity in more than 1000 kinds of human tumor cells, which can be used to assess cancer gene mutation and the influence of different gene expression on the anti-cancer drug sensitivity. In this study, human tumor cells in the database were grouped into Cluster A and ClusterB by using the clustering

method in this study based on the screened 5 macrophage-related hub genes, and the sensitivity differences of the two groups to anticancer drugs were evaluated. The prediction accuracy of this method was verified by 10-fold cross validation, and drug sensitivity was estimated by IC50.IC50 is the ratio of apoptotic cells to all cells corresponding to a drug concentration of 50%. The lower the IC50, the more sensitive to the drug. The aim of this study was to further explore the potential of hub genes screened in this study as therapeutic biomarkers for predicting drug responses by evaluating the differences in response to anticancer drugs between the two clusters.

## RESULTS

### Data downloading and collection

A total of 85 TNBC patients from the TCGA dataset were included in this study, with a mean age of 53.87 years and a standard deviation of 12.19 years (Table 1). The other part of the validation set was taken from the GEO database. The dataset GSE103091 included 107 patients with TNBC, with a mean age of 56.97 years, and a standard deviation of 12.8 years.Dataset GSE76124 included 195 patients with TNBC, with a mean age of 54.58 years, and a standard deviation of 12.68 years. The relevant patient information is listed in Tables S1 and S2. GSE118389 and GSE161529 are single-cell datasets in GEO database, which included 14 triple-negative breast cancer tissue samples with a total of 45,710 cells.

### Immune infiltration analysis and WGCNA screening of macrophage-related genes in TNBC

The ssGSEA, EPIC, MCP Counter, QUANTISEQ, Cibersort, and XCELL (giving a total of six types of immune infiltration analysis method) were used to study the immune components of 85 TNBC patients in the TCGA database. At the same time, we performed WGCNA analysis on the data to identify the modules in TNBC that are associated with certain immune cell infiltrates. We found when we set the soft threshold to 6, it conforms to the scale-free characteristic of biological networks, so we use $\beta = 6$ to construct a weighted network (Fig. 2A). Secondly, average linkage hierarchical clustering, based on TOM differences, dynamic tree pruning, and merging to identify modules, we obtained a total of 26 meaningful modules marked with different colors (Fig. 2B). Then, all the modules analyzed in WGCNA were correlated with the immune cell score results obtained by different immune infiltration analysis methods (Fig. 2C; Figs. S1–S5). We found that the blue module was positively correlated with macrophages in all six algorithms (Fig. 2D). Then, 794 genes in the blue module were subjected to enrichment analysis. GO analysis showed that they were mainly enriched in biological processes related to macrophages and neutrophils. KEGG and Reactome analyses proved that genes in the blue module were mainly enriched in immune-related pathways (Fig. 2E). So far, we have found a group of stable macrophage-related genes in TNBC, and their biological functions are correlated with tumor immunity to a certain extent.

### Screening of hub genes based on cluster analysis

The further to explore the characteristics of genes in the blue module, we performed a $k$-value based consistent clustering based on the expression of 794 genes involved in the

**Table 1 The basic characteristics of the patients in TCGA-TNBC.**

| Characteristic | Freq |
| --- | --- |
| AJCC_Status, No. (%) | |
| StageI | 14 (16%) |
| StageII | 53 (62%) |
| StageIII | 17 (20%) |
| StageIV | 1 (1%) |
| Age (mean ± SD) | 53.87 ± 12.19 |
| PAM50, No. (%) | |
| Basal | 73 (86%) |
| Her2 | 6 (7%) |
| LumA | 3 (4%) |
| LumB | 1 (1%) |
| Normal | 2 (2%) |
| Stage_M, No. (%) | |
| M0 | 84 (99%) |
| M1 | 1 (1%) |
| Stage_N, No. (%) | |
| N0 | 51 (60%) |
| N1 | 22 (26%) |
| N2 | 7 (8%) |
| N3 | 5 (6%) |
| Stage_T, No. (%) | |
| T1 | 18 (21%) |
| T2 | 56 (66%) |
| T3 | 9 (11%) |
| T4 | 2 (2%) |

blue module. The cumulative distribution function (CDF) graph shows the cumulative distribution function for $2 \leq k \leq 6$ (Fig. S6). We selected $k = 2$ as the optimal parameter, and divided triple negative breast cancer patients into two groups: Clusters I and II (Fig. S7). Further, survival analysis revealed significant differences in OS between patients in Clusters I and II ($P = 0.0038$; Fig. 3A), thus, we inferred that macrophage-related genes in the blue module could affect the OS of patients with TNBC through immune-related pathways.

To screen hub genes in the module, genes with a GS $P$ value of less than 0.001 were selected from the blue module of WGCNA for univariate KM analysis. When the $P$ value in the KM analysis was less than 0.05, 12 candidate genes could be screened out. Furthermore, through the repeated Lasso analysis (1,000 times, Fig. 3B), we finally selected five hub genes: CD79A, CXCL13, IGLL5, LHFPL2, and PLEKHF1. We identified it as the most stable macrophage-related gene associated with prognosis in TNBC.

To explore the value of these five hub genes, according to the expression of these five genes, we applied the consistency cluster analysis based on the $k$ value, and selected $k = 2$ as the optimal parameter according to the cumulative distribution function (Fig. 3C). According to the expression of these five hub genes, TNBC patients in TCGA were

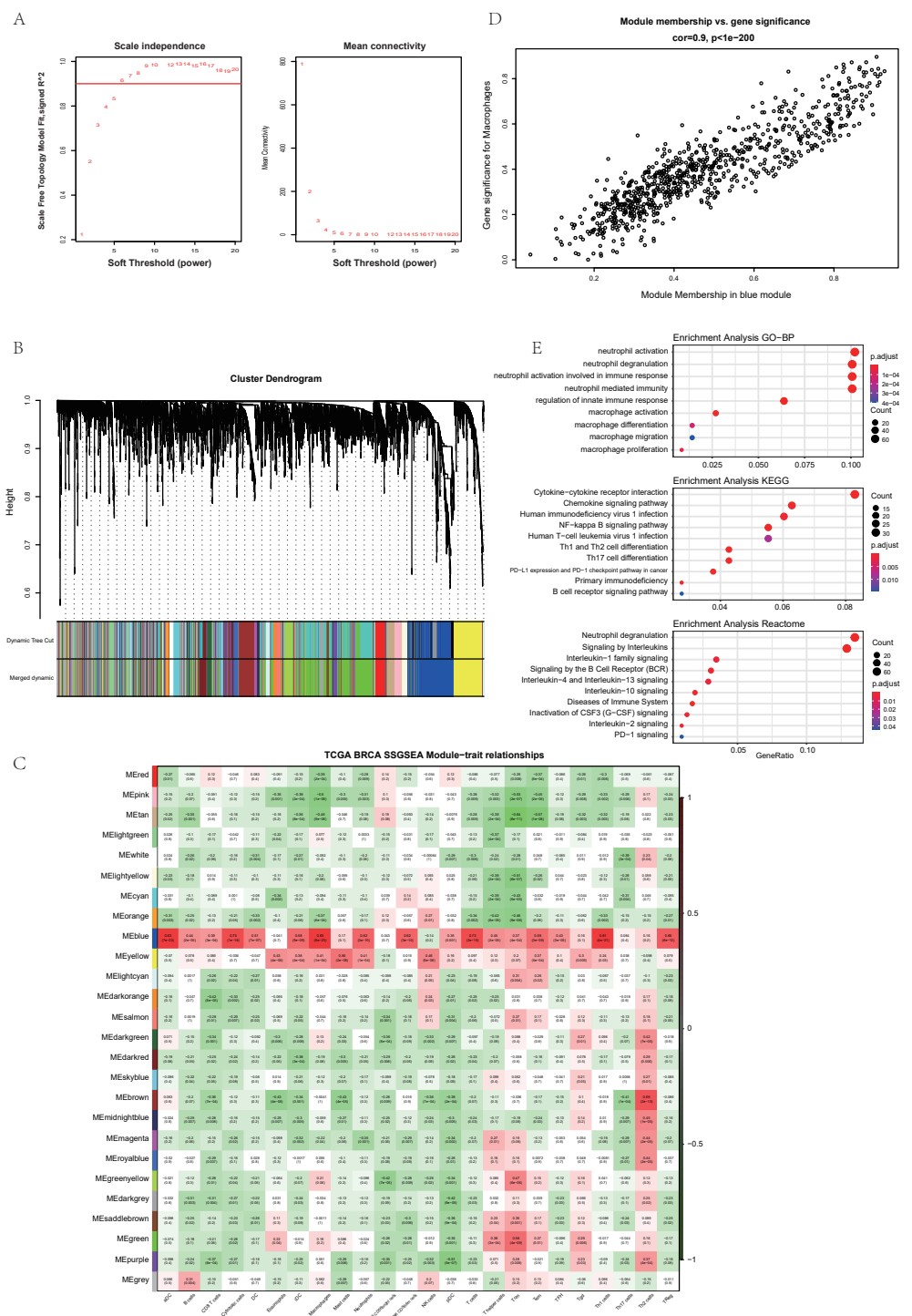

**Figure 2 Immune infiltration analysis and WGCNA module screening of macrophage related genes in TNBC.** (A) Soft threshold selection in the WGCNA network analysis. (B) Gene distribution in the WGCNA network analysis. (C) Correlation analysis of different modules in WGCNA and SSGSEA immune cell score. (D) Correlation analysis of blue module in WGCNA and macrophage score in TNBC (E) Enrichment analysis for the genes in macrophage related module.

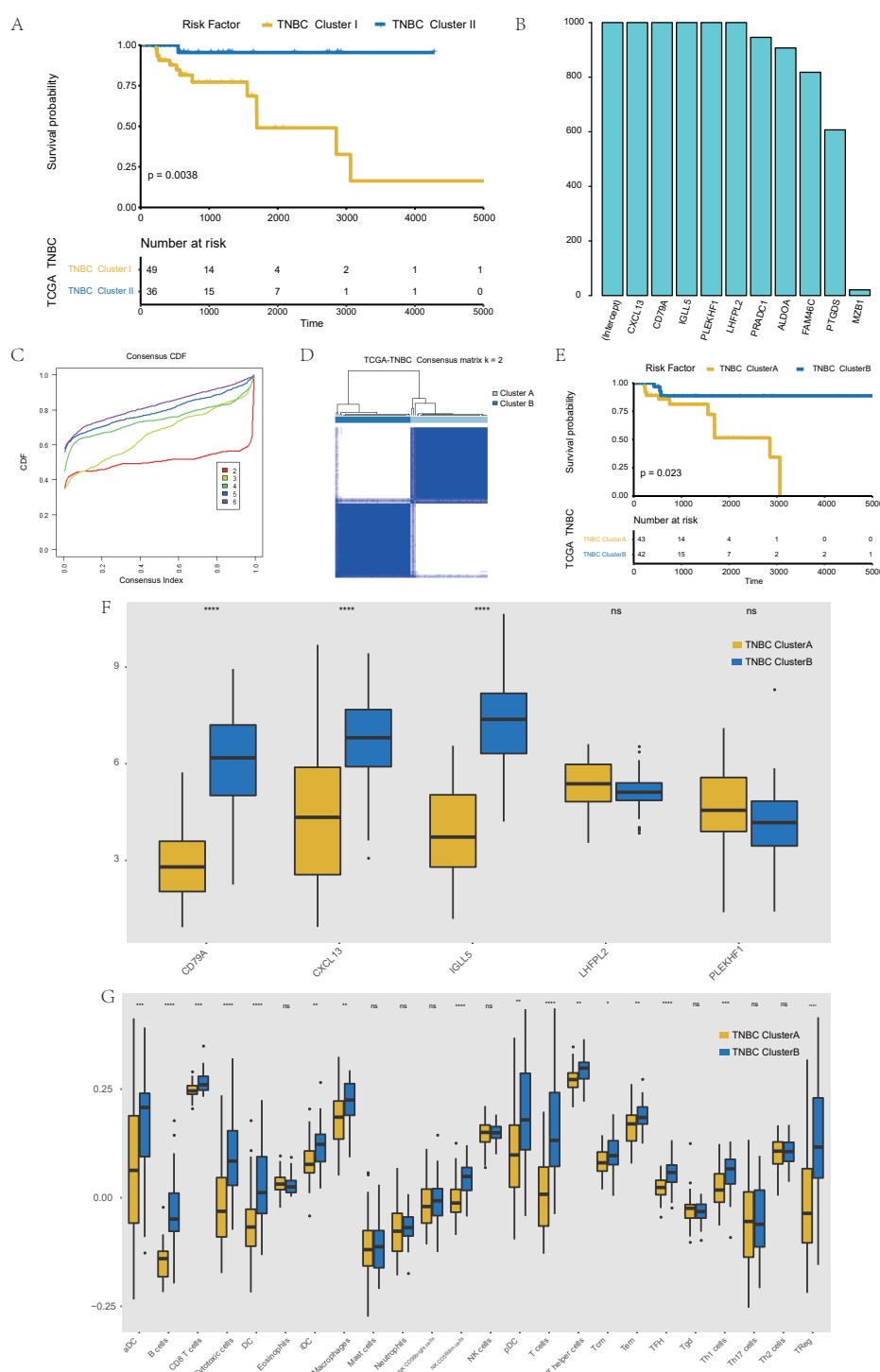

**Figure 3  Screening of hub genes and their related expressions.** (A) Relationship between different groups (Cluster I and Cluster II) and OS in TNBC patients. (B) 1000 Lasso Cox regression screening of macrophage-related hub genes. (C) The cumulative distribution (continued on next page...)
**Figure 3 (…continued)**
function of consensus clustering of TNBC patients in TCGA database based on the consensus clustering of five hub genes when k = 2-6 (D) Sample clustering heatmap of TNBC patients in the TCGA database at $k = 2$ based on consensus clustering of 5 hub genes. (E) Associations between hub genes-based consistent clustering grouping (ClusterA and ClusterB) and OS in TNBC patients. (F) Differences in the expression levels of five hub genes between different clusters (ClusterA and ClusterB) (G) Difference of macrophage scores between different clusters (ClusterA and ClusterB).

divided into two groups: Clusters A and B (Fig. 3D). Survival analysis showed that the OS of Cluster A was significantly lower than that of Cluster B, and the difference was statistically significant ($P = 0.023$) (Fig. 3E). In addition, we explored differences in hub gene expression and immune scores between Clusters A and B (Fig. 3F). Among the five hub genes, three genes showed significant differences in their levels of expression in the two clusters ($P < 0.001$), and the results (Fig. 3G) showed that the difference in macrophage scores between Clusters A and B was statistically significant ($P < 0.01$).

## Validation of key genes that have been screened

To verify the stability of consistency clustering based on the $k$ value in the classification of TNBC patients in TCGA, another classification method (PCoA analysis), was selected for verification, and the results are shown in Fig. 4A. To prove the accuracy of data analysis in the TCGA database, we chose another two datasets (GSE103091 and GSE76124) to prove that changes in the dataset would not affect our conclusion. In GSE103091, the data of 107 TNBC patients were selected for clustering analysis according to the expression of the five hub genes. The TNBC patients in GSE103091 were divided into two groups: Clusters C and D. Based on the consistency clustering of the $k$ values, the results are as shown in Figs. 4B–4C. Survival analysis between the two groups showed a statistically significant difference in OS (Fig. 4D, $p = 0.0017$). .In addition, the results of PCoA cluster analysis are illustrated in Fig. 4E. Three of the five hub genes were expressed differently in the two clusters ($P < 0.05$) (Fig. 4H). The results obtained in GSE103091 are consistent with those obtained in TCGA. Similarly, TNBC patients in GSE76124 can be divided into two groups: Clusters E and F, and we applied cluster analysis results as demonstrated in Figs. 4F and 4G; five hub gene expression in the two clusters have obvious differences (Fig. 4I), but due to the dataset not providing survival information of patients, we only verified the grouping situation, since then, we proved that in a different dataset the stability and the accuracy of the above key genetic screening were maintained.

## Development of patient grouping criteria based on the hub gene

The score based on five hub genes constructed by multivariate Cox regression analysis is as follows:

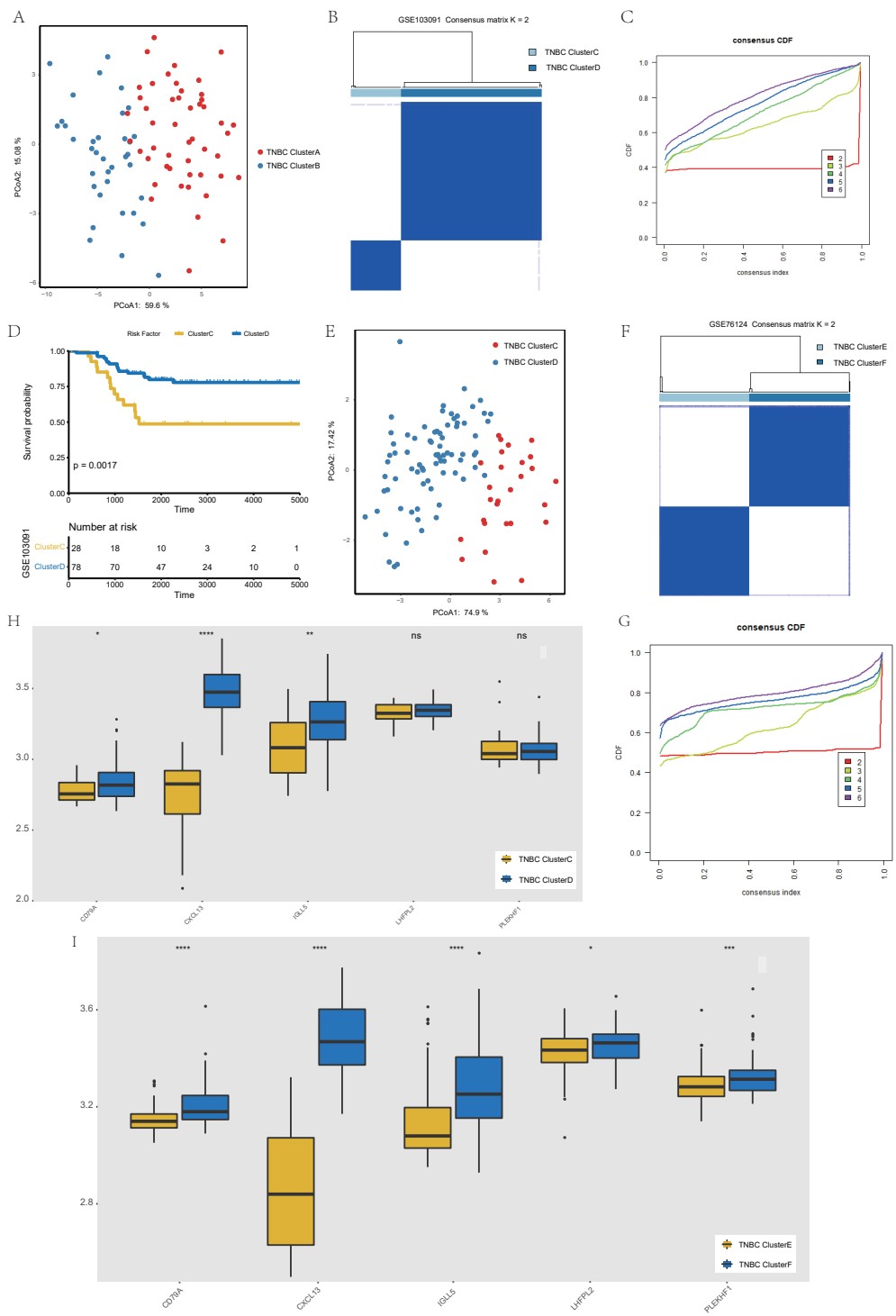

**Figure 4** **Validation of previously screened hub genes.** (A) Principal Coordinate Analysis (PCoA) scatter plot of TNBC patients based on five hub genes in TCGA dataset. (B) Sample clustering heatmap of TNBC patients (continued on next page...)

**Figure 4 (...continued)**
in the GSE103091 dataset at $k = 2$ based on consensus clustering of five hub genes. (C) The cumulative distribution function of consensus clustering of TNBC patients in the GSE103091 dataset based on the consensus clustering of five hub genes when k = 2-6. (D) Associations between hub genes-based consistent clustering grouping(ClusterC and ClusterD) and OS in TNBC patients. (E) Principal Coordinate Analysis (PCoA) scatter plot of TNBC patients based on five hub genes in the GSE103091 dataset. (F) Sample clustering heatmap of TNBC patients in the GSE76124 dataset at $k = 2$ based on consensus clustering of five hub genes (G) The cumulative distribution function of consensus clustering of TNBC patients in the GSE76124 dataset based on the consensus clustering of five hub genes when k = 2-6. (H) Differences in the expression levels of five hub genes between different clusters (ClusterC and ClusterD) (I) Differences in the expression levels of five hub genes between different clusters (ClusterE and ClusterF).

$$Score = CXCL13\ Exp * 48.01469 + CD79A\ Exp * 74.04190 + IGLL5\ Exp$$

$$* 83.01726 - PLEKHF1\ Exp * 153.23944 - LHFPL2\ Exp * 82.86847.$$

According to the previous clustering analysis results, the cut-off value was $-42$. 021.TCGA-TNBC was used as the training dataset and GSE103091 and GSE76124 as the verification dataset. The accuracy of the cut-off value was verified by area under the ROC curve (AUC) analysis, with AUC values respectively of 1.000 (Fig. 5A), 1.000 (Fig. 5B), and 0.914 (Fig. 5C). It is proved that this cut-off value can accurately divide TNBC into two clusters.

## Verification of correlation between HUB gene and macrophages based on single cell sequencing

A total of 45,710 cells in GSE118389 and GSE161529 were preliminarily screened for quality control and normalization (Fig. 6A). These cells were mainly divided into seven groups by T-SNE method and cell types were annotated by Cellmarker database, as follows: Epithelialcell, Macrophage, Endothelial_cells Tissue_stem_cells, Fibroblasts, B cell and T cell. (Fig. 6B). Cell composition proportion of 14 triple-negative breast cancer samples is shown in Fig. 6C. The number distribution of each cell type in all 45710 cells is shown in Fig. 6D. Figure 6E shows the expression of the five hub genes screened in various cell types, and the results show that: CD79A and IGLL5 are mainly expressed in B cells, CXCL13 is mainly expressed in T cells, LHFPL2 is mainly expressed in macrophages, and PLEKHF1 is slightly expressed in various cells, which is not clear yet.We found that these five hub genes were mainly expressed in T cells, B cells and macrophages, so we used CellChat method to analyze cell communication, and Figs. 6F–6G shows the interaction between these three cells and genes mainly expressed in these cells. One of the most valuable results was shown in Fig. 6H, where we found that T cells and B cells could simultaneously act on macrophages mediated by CD45 signaling pathway. We hypothesized that CXCL13 expressed in T cells and CD79A and IGLL5 expressed in B cells could affect the expression of LHFPL2 gene through the CD45 signaling pathway, acting on macrophages, thus jointly playing a role in tumor regulation.

## Construction and test of prognosis model based on hub genes

Univariate Cox regression and multivariate Cox regression analyses were used to screen the clinical pathological parameters related to the prognosis of patients. Results of univariate

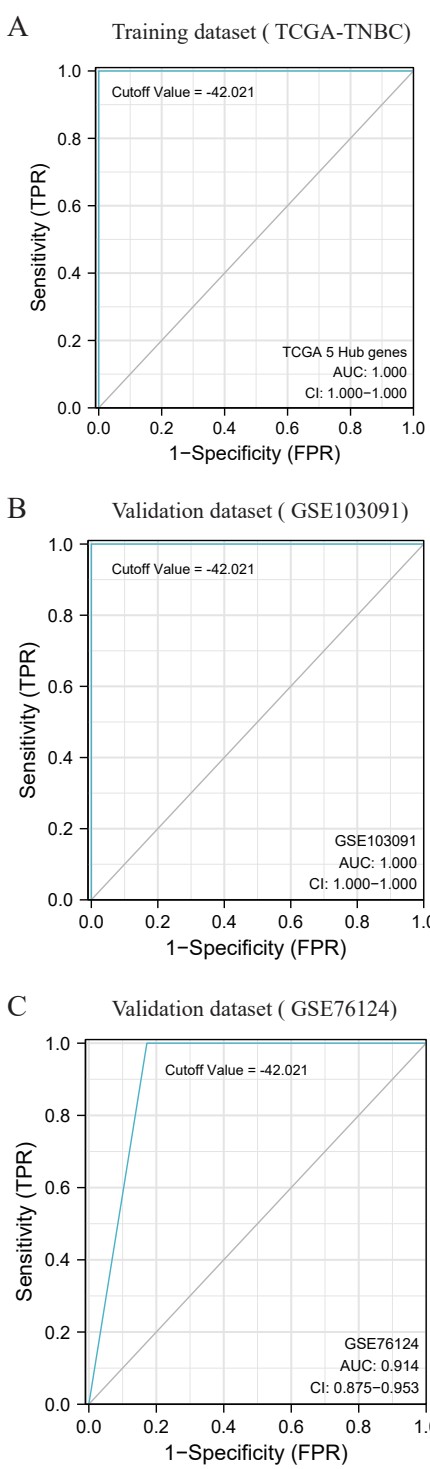

**Figure 5  Accuracy test of grouping criteria for TNBC patients.** (A) ROC curve of training dataset (TCGA-TNBC) (B) ROC curve of validation dataset (GSE103091) (C) ROC curve of validation dataset (GSE76124).

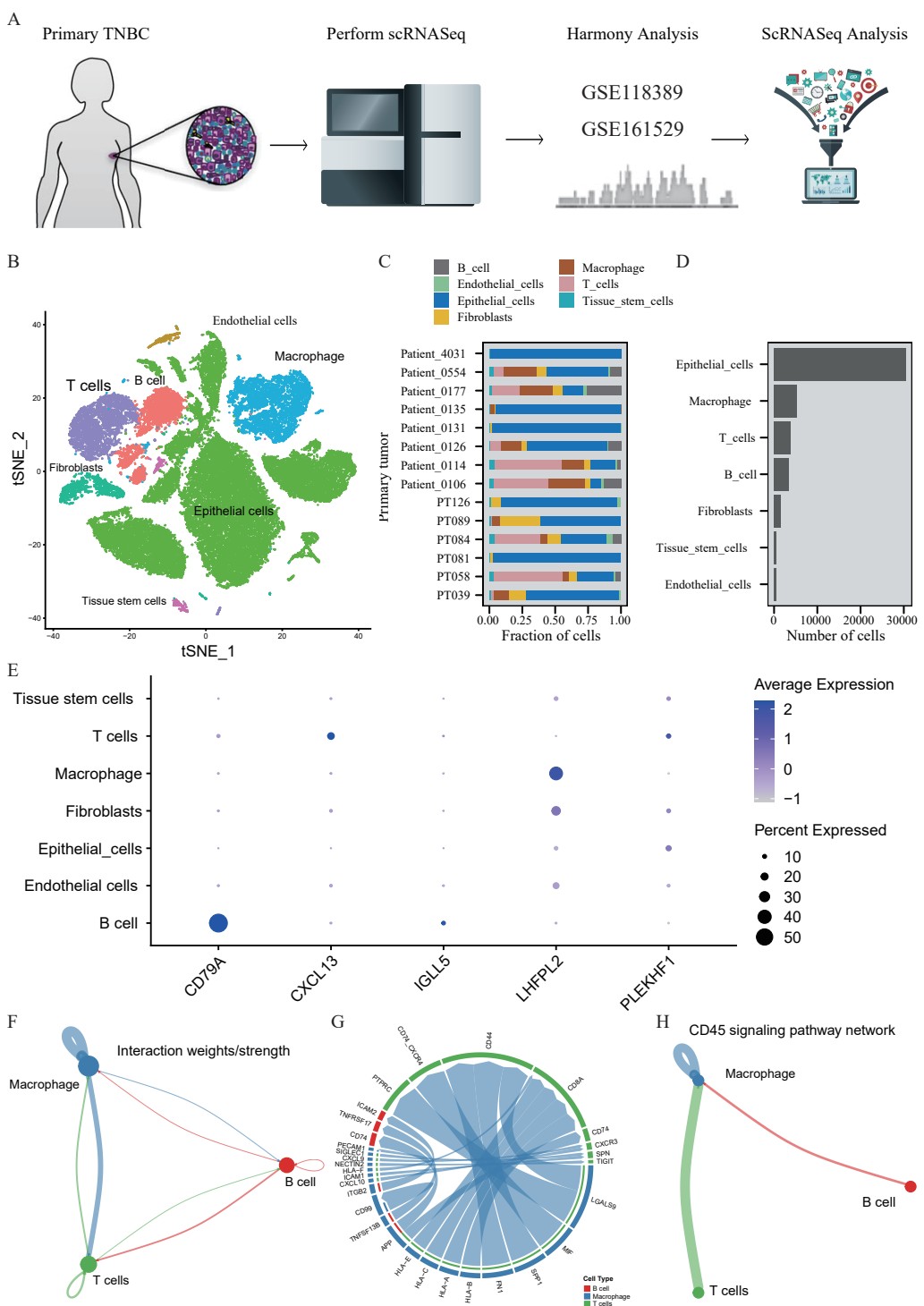

**Figure 6** **Verification of interaction between five hub genes in tumor microenvironment with TNBC scRNA-seq.** (A) Workflow diagram showing the collection and processing of 14 primary TNBC tumors from two published GEO Datasets for scRNA-seq; (B) t-SNE plots of cells (continued on next page...)

**Figure 6 (…continued)**
from the 14 samples profiled in this study, with each cell color coded to indicate the associated cell types. (C) The fraction of cells originating from each patient; (D) The number of cells originating from each patient; (E) The five hub genes expression in different cell types (color represents the average value of gene expression in cell types; the size of the dot represents the proportion of cells expressing the gene) (F) Capacity for intercellular communication between T cells, B cells and Macrophage cells. (G) Detailed view of the ligands expressed by each major cell type and the cells expressing the cognate receptors primed to receive the signal. (H) Signaling pathway for intercellular communication between T cells, B cells and Macrophage cells (Kyoto Encyclopedia of Genes and Genomes).

**Table 2   The correlation of the clinicopathological parameters with patients's overall survival.**

| characteristics | Univariate Cox | | | Multivariate Cox | | |
|---|---|---|---|---|---|---|
| | **HR** | **95% CI** | ***P*-value** | **HR** | **95% CI** | ***P*-value** |
| Age | 1.02 | 0.98–1.07 | 0.298 | NA | NA | NA |
| AJCC_Status | 4.77 | 1.97–11.54 | 0.001 | 1.69 | 0.33–8.82 | 0.531 |
| Group | 0.25 | 0.07–0.9 | 0.035 | 0.16 | 0.04–0.65 | 0.01 |
| PAM50 | 0.96 | 0.47–1.93 | 0.899 | NA | NA | NA |
| Stage_M | 36.15 | 3.28–398.72 | 0.003 | 0.81 | 0.04-14.89 | 0.888 |
| Stage_N | 3.68 | 1.97–6.9 | 0 | 3.6 | 1.19–10.93 | 0.023 |
| Stage_T | 1.89 | 0.86–4.14 | 0.112 | NA | NA | NA |

Cox regression analysis with $P$ values less than 0.2 were included as alternative variables in multivariate Cox regression analysis (Table 2). The results showed that five hub genes groups and N staging were prognostic risk factors. We plotted a histogram based on the grouping results of the five hub genes and N staging as prognostic factors (Fig. 7A). Time-dependent ROC curves were used to study the predictive power of the prognostic model, and the AUC values at one year, three years, five years, seven years, and ten years were 0.77, 0.58, 0.71, 0.75, and 0.96 (Fig. 7B). The larger the AUC is, the better the discriminant ability of the prediction model. From the analysis results of this study, the degree of differentiation at 3 years is close to the average, but those at 1 year, 5 years, 7 years, and 10 years is very good. Calibration analysis was conducted for the prognostic model. The results at one year, three years, and five years are shown in Figs. 7C–7E. The prediction results of the model proposed in this study closely matched the real value without statistically significant differences, showing that this prognostic model had high prediction accuracy.

We validated the clinical benefit of the predictive model using a clinical decision curve, and the results showed that patients using the predictive model had a higher net clinical benefit than patients treated directly or not treated at all (Fig. 7F).

## Drug sensitivity analysis based on GDSC
Based on GDSC database, the drug sensitivity of TNBC patients in Clusters A and B was studied to compare the differences of drug sensitivity among different clusters. Notably, we found no difference in the sensitivity of Clusters A and B to the chemotherapeutic drug gemcitabine (Fig. 8A), while there was a significant difference in the sensitivity to vinorelbine (Fig. 8B). Among them, patients in Cluster A were more sensitive to vinorelbine

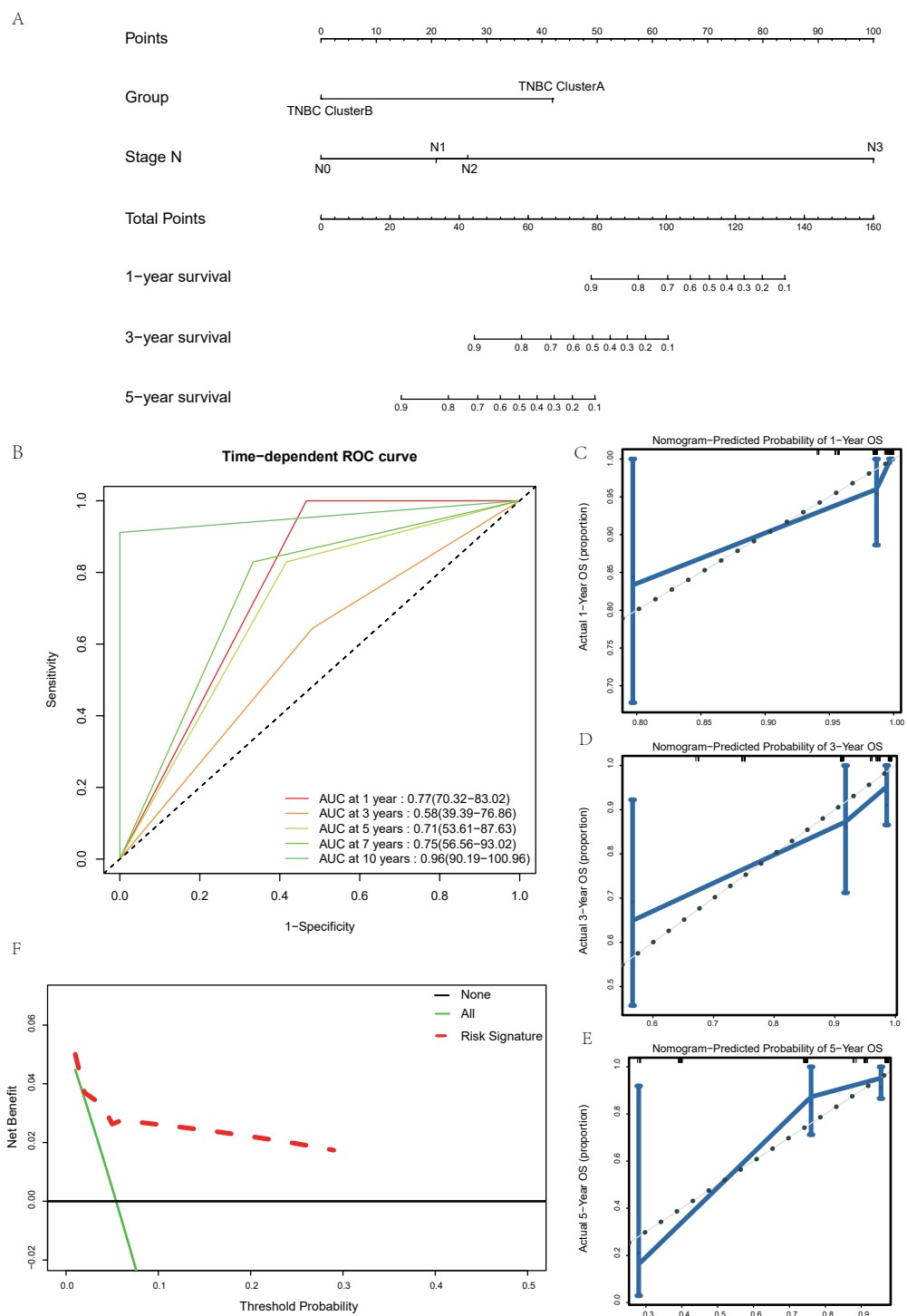

**Figure 7 Construction and test of prognosis model based on hub genes.** (A) The nomogram of prognosis model based on the five hub gene groupings of TNBC. (B) Time dependent ROC curve of the TNBC prognostic model. (C) The one-year calibration curve of the TNBC prognostic model. (D) The three-year calibration curve of the TNBC prognostic model. (E) The five-year calibration curve of the TNBC prognostic model. (F) Clinical decision curve of the TNBC prognostic model.

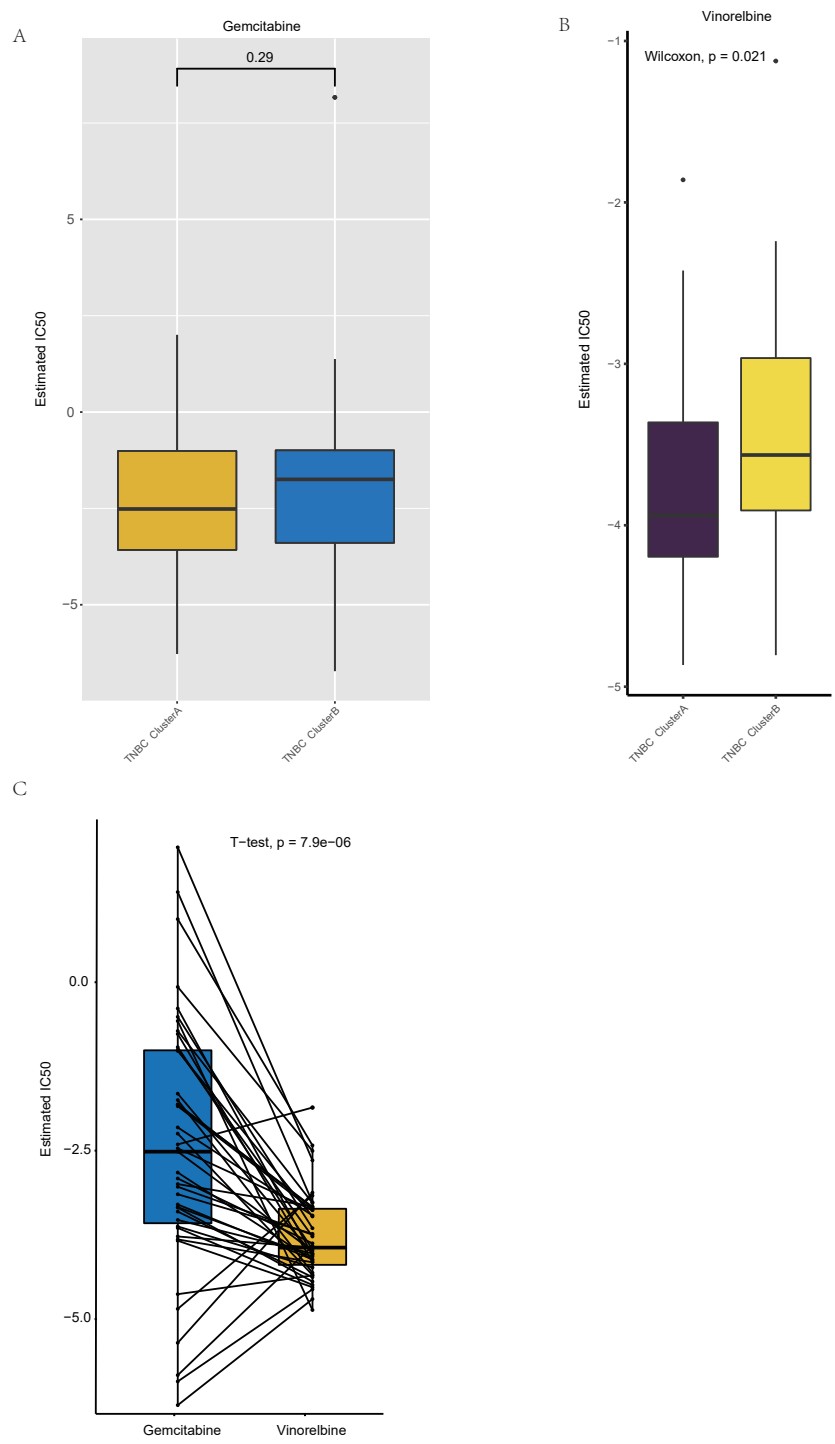

**Figure 8** **Drug sensitivity analysis of ClusterA and ClusterB based on GDSC.** (A) Comparison of sensitivity of ClusterA and ClusterB to Gemcitabine. (B) Comparison of sensitivity of ClusterA and ClusterB to Vinorelbine. (C) Comparison of sensitivity of TNBC patients in ClusterA to Gemcitabine and Vinorelbine.

(IC50 values were lower). Both gemcitabine and Vinorelbine can be used as monotherapy or in combination with platinum for the treatment of patients with advanced breast cancer who have failed to respond to anthracene and taxane therapy (*Seidman, 2001*). Further, the paired $t$-test was used to reveal the sensitivity differences between gemcitabine and Vinorelbine in patients in Cluster A. It was found that patients in Cluster A had a higher sensitivity to Vinorelbine, with a statistical difference (Fig. 8C). Previous analysis has shown that Cluster A has a worse prognosis than Cluster B, and based on the above analysis, we speculated that Vinorelbine could be preferred for further treatment in this group of patients with poor prognosis after the onset of anthracene and taxane drug resistance.

## DISCUSSION

The heterogeneity of TNBC is very strong (*Burstein et al., 2015*; *Lehmann et al., 2011*) and in 2019, staff at Fudan University, China team mapped the world's biggest multiple-omics map of TNBC (The TNBC Fudan classification (2019)). Researchers tried many methods of classification to distinguish the different TNBCs, aimed at making accurate target therapy for TNBC possible. In recent years, some tumors have gradually benefited from immunotherapy. As the type of breast cancer with the strongest immunogenicity, we believe that distinguishing TNBC from the perspective of a certain tumor associated immune cell is also a new idea worth exploring. In this study, through six types of immune cells composition analysis methods, WGCNA expression analysis, two kinds of clustering analysis, LASSO analysis, and other methods, screened out five macrophage-related hub genes affecting the prognosis of TNBC. TNBC was divided into two groups based on the different expression levels of the five macrophage-related genes and set the grouping criteria. The analysis of prognosis and drug sensitivity showed that there were significant differences in prognosis and sensitivity to gemcitabine between the two groups. Although this study only classified TNBC from a certain perspective, it has certain reference value for prognosis and clinical individualized medication.

The exploration process used in the present study was both rigorous and reliable. The module genes positively related to macrophages were selected based on the analysis results obtained by six immune infiltration analysis methods. In addition, those macrophages thus analyzed have been demonstrated in many studies to be closely related to the inflammatory response, immune regulation, invasion and metastasis of tumors, and other biological processes (*Hanahan & Coussens, 2012*; *Zhang et al., 2010*). In the general sense, macrophages can be divided into an M1 type with a tumor suppressive effect and an M2 type with a tumor promoting effect (*Hu et al., 2016*). Under different conditions, the two subtypes of macrophages can transform each other to perform different functions. Among the six immunoinfiltration analysis methods used in this study, three could distinguish M1-type macrophages from M2-type macrophages. We believe that the genes in the module in this study are more strongly correlated with M2-type macrophages for two reasons. First, existing studies have shown that TAM plays a similar role influencing the tumor microenvironment to favor M2-type macrophages and can produce many anti-inflammatory factors to promote tumor development. Breast tumor cells can secrete

cytokines that promote the polarization of macrophages to M2 type, and compared with other types, TNBC can secrete more granulocyte colony stimulating factor (G-CSF) to promote the polarization of M1 to M2 type (*Hollmén et al.*). Secondly, GO-BP enrichment analysis showed that this module gene was enriched in immune-related pathways, and the neutrophil-related pathways contained in the results were also worthy of consideration. Studies have shown that, in the tumor microenvironment (TME), both macrophages and neutrophils are involved in the regulation of immune response in both cancer and inflammation, and there may be different degrees of interaction between the two (*Galdiero et al., 2013*). Mature neutrophils release IL-8 and TNF-α to activate and recruit macrophages, providing signals for macrophage activation and maturation (*Kumar & Sharma, 2010*). OSM secreted by neutrophils can promote the polarization of macrophages towards the M2 type. TGF-β released by tumor-associated neutrophils (TAN) can also regulate the polarization of macrophages and promote their differentiation into a pro-tumor phenotype (M2 type) (*Wu et al., 2019*). The enrichment analysis results not only confirmed the correlation between the genes in this module and the immune function, but also allowed the inference that the genes in this module are more strongly correlated with M2-type macrophages. In the genes of macrophage related co-expression module, through single-factor KM analysis, 1,000 cycles of Lasso analysis, and so on, five hub genes were strictly screened out, respectively: CD79A, CXCL13, IGLL5, LHF-PL2, and PLEKHF1.

Among these five genes, CXCL3 and PLEKHF1 have been reported in breast cancer related studies, among which CXCL13 has been reported to be highly-expressed in breast cancer and significantly correlated with lymph node metastasis, distant metastasis, and disease staging (*Jiang et al., 2020*). In addition, it is also associated with poor prognosis of breast cancer and TIL density (*Razis et al., 2020*). PleKHF1 has only been reported to be highly-expressed in 19q12-amplified ER-negative breast cancer (*Natrajan et al., 2012*). CD79 has been extensively studied in hematological tumors and has been reported to be more positive in elderly Hodgkin's lymphoma patients with poor prognosis (*Sakatani et al., 2020*); however, IGLL5 and LHFPL2 are less often studied in tumors and their functions remain unknown.

Based on the five hub genes, TNBC patients were divided into two groups (Clusters A and B), by consistent clustering method, the results showed that the prognosis of the two groups was significantly different. Therefore, we concluded that the five hub genes were related to the macrophages of TNBC patients and could affect the OS of patients through immune-related pathways. Then, to demonstrate the stability of the algorithm, we chose another clustering method, PCoA. To determine the stability of the dataset, we selected GEO database datasets GSE103091 and GSE76124, and both obtained results consistent with the previous analysis. Next, we formulated the inclusion criteria based on the previous grouping results and selected the appropriate cut-off value, so that this grouping based on five hub genes could be better applied in practical clinical applications. The good results of two datasets, GSE103091 and GSE76124, as validation sets prove the accuracy and applicability of the present research.

In order to prove the correlation between these five HUB genes and macrophages, we also applied single-cell data for analysis. All cells in the TNBC sample were sorted and

annotated, and it was found that these five genes were mainly expressed in T cells, B cells and macrophages. By analyzing the intercellular communication of these three cells, it was found that T cells and B cells could simultaneously act on macrophages mediated by CD45 signaling pathway. Therefore, it is reasonable to speculate that genes expressed in T and B cells may modulate genes expressed in macrophages through the CD45 signaling pathway, thus jointly playing a role in tumor regulation. In addition, it has been reported that CD45 is the receptor of CD24D, and the immunomodulatory peptide CD24D can reverse the immunosuppression induced by TNBC by activating the CD45 signaling pathway, and participate in the biological process of immunomodulatory tumor cell killing. CD45 may be the target of TNBC immune response recovery (*Raiter et al., 2021*). In conclusion, we believe that even though these five hub genes are not all expressed on macrophages, they can all play a role through macrophages, which also demonstrates the accuracy of them as macrophage-related genes.

By examining the clinical application value of these five hub genes, we built a prognostic model with five gene grouping and N staging as independent prognostic factors, and demonstrated the good predictive ability, high accuracy, and patient benefits of this model respectively through ROC curve, calibration, and decision curve analyses. In the ROC curve, compared with the AUC value within three years, the AUC value beyond five years is larger and the prediction effect is better, indicating that this model provides better predictions of long-term survival.

In addition to the prediction of patient survival, we further explored the differences in the sensitivity of the two groups of people to different chemotherapy drugs. Previous studies have reported that macrophages can participate in mediating chemotherapy drug resistance and thus play a certain biological role in the occurrence and development of breast cancer. CSF1 is a factor that can promote the formation of tumor-associated macrophages. In a mouse transplantation model based on the MCF-7 cell line, blocking CSF1 expression improved the sensitivity of cancer cells to chemotherapy drugs and reversed chemotherapy resistance (*Liverani et al., 2014*). Granulocyte-macrophage colony stimulating factor (GM-CSF) can also stimulate apoptosis of epirubicin-resistant breast cancer cell lines by promoting the transformation of M2 type macrophages to M1 type macrophages (*Lei et al., 2018*). In this study, no difference was found between Cluster A and Cluster B in terms of drug sensitivity to gemcitabine, and patients in Cluster A were more sensitive to Vinorelbine than those in Cluster B. Patients in Cluster A were also more sensitive to Vinorelbine than gemcitabine. Gemcitabine and vinorelbine are common chemotherapeutic agents used on advanced TNBC patients, and they are used as a relief treatment for anthracene and taxane resistance. The patients in Cluster A were confirmed to have poor prognosis in previous analysis, therefore, we concluded that Vinorelbine should be selected as the first chemotherapy drug when patients of Cluster A developed resistance to anthracene and taxanes. The five hub genes screened out in this study may be involved in regulating the drug resistance of TNBC to Vinorelbine, but the specific mechanism thereof warrants further exploration. Taken together, we hypothesized that these five genes could be used as biomarkers for clinical drug decision making after anthracene and taxane resistance in advanced TNBC patients.

In general, five hub genes associated with macrophages in TNBC were screened out in this study, and the OS of patients could be affected through immune-related pathways. Based on these five genes, we constructed a prognostic model for predicting survival in patients with TNBC. In addition, we concluded that these five genes may be involved in vinorelbine resistance in TNBC patients, and the TNBC classification based on these five hub genes could be used to assist drug-related decision-making in TNBC patients after onset of anthracene and taxane chemotherapeutic resistance. There remain some limitations to the present research, mainly reflected in the fact that the study was based on the exploration of public databases, and experimental verification is still needed; only the influence of hub gene on the functional level of TNBC has been investigated, and the specific regulatory mechanism therein remains to be explored.

In general, we believe that from the perspective of immune infiltration, it is novel to select specific immune cell - macrophage-related hub genes to group TNBC, evaluate prognosis, and guide clinical chemotherapy medication decision-making. The logic behind this study was rigorous; however, there remain some limitations in this work: this study was based on the bioinformatics analysis of entries in public databases and lacks the validation of test evidence from clinical samples, therefore, TNBC patients will be included in the follow-up study to verify the results of this study from a clinical perspective.

## CONCLUSIONS

In summary, six immune infiltration analysis methods and two clustering methods were applied in this study. Based on two public databases (GEO and TCGA), five hub genes related to macrophages were screened out, and a prognostic model for predicting the survival of TNBC patients was established. Drug-sensitivity analysis in the GDSC database showed that these five genes were associated with gemcitabine resistance, and we inferred that they could serve as biomarkers for clinical drug decision-making following the onset of anthracene resistance and taxane resistance in patients with advanced TNBC.

### Funding
The authors received no funding for this work.

### Competing Interests
The authors declare there are no competing interests.

### Author Contributions
- Peng Su performed the experiments, analyzed the data, prepared figures and/or tables, authored or reviewed drafts of the paper, and approved the final draft.
- Ziqi Peng and Boyang Xu analyzed the data, prepared figures and/or tables, authored or reviewed drafts of the paper, and approved the final draft.
- Bowen Yang conceived and designed the experiments, performed the experiments, analyzed the data, authored or reviewed drafts of the paper, and approved the final draft.

- Feng Jin conceived and designed the experiments, authored or reviewed drafts of the paper, and approved the final draft.

## Data Availability

The data is available at TCGA-BRCA (https://portal.gdc.cancer.gov/ and NCBI GEO: GSE103091, GSE76124.

The analysis code is available at GitHub:

https://github.com/HopeStar2018/Macrophages-Related-Gene-Signature-to-Predict-Overall-Survival-in-Patients-with-TNBC.git

## Supplemental Information

Supplemental information for this article can be found online at http://dx.doi.org/10.7717/peerj.12383#supplemental-information.

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
