# Peer review of "Establishment and validation of an individualized macrophage-related gene signature to predict overall survival in patients with triple negative breast cancer"

_PeerJ, doi:10.7717/peerj.12383_

## Round 0.1 · original submission · Major Revisions

The subject explored in this study is a relevant one - personalized medicine is an upcoming field that will benefit patients based on their specific profile. The study shows a novelty that could suggest which subclass of patients will fail for gemcitabine treatment and which subclass could benefit from vinorelbine treatment. That said, the manuscript needs improvement in several aspects to be published in the PeerJ. The English must be carefully revised. It is important to answer the reviewer's questions and ensure the improvements on the analysis have been performed.

Reviewer 1 ·

Basic reporting

The manuscript needs heavy editing and improvement with respect to the use of professional English usage and sentence constructions. It made the flow of the paper very difficult to follow.
Even though the authors discussed TAMs in the introduction thoroughly, the authors in their introduction and discussion mentioned that they analyzed them based on "certain immune cells". Since this paper talks about TAMs in the introduction and does analysis based on macrophages it is very important to highlight what was the grouping criteria. Thus, it didn't make a strong rationale for their study.
Figures were not legible even at 200% zoom which makes it difficult to follow with the text.
Figure 3 needs to be completely changed as the labels do not match the text in the manuscript.

Experimental design

The authors did extensive analysis and validated their observations with multiple data sets and programs. This was one of the strongest points of this paper. I commend the detailed analysis and flow of the research rationale.
Methods are described in detail. Authors are suggested to add a Schematic of the Study Design and analysis used in the paper. It will be easier for readers to follow.

Validity of the findings

Personalized medicine is an upcoming field and providing immunotherapies to patients based on subclasses could be really helpful. The authors have done a good job in showing the novelty and robustness of their subclass analysis that could suggest which subclass of patients will fail for gemcitabine treatment and which subclass could benefit from Vinorelbine treatment.
The weakness of the paper is the disconnect between their discussion on TAMs whereas in the paper they described that they were distinguishing TNBC from the perspective of a "certain immune cell" without providing much context on what is this certain immune cell?

Additional comments

The authors provided a prognostic analysis on a TNBC patient's overall survival and resistance to gemcitabine with respect to immune cell infiltration. Such analysis is important as the tumor microenvironment has been shown to be involved in chemotherapy resistance.
All corrections that are required to be made in the paper could be found in the annotated version.

Annotated reviews are not available for download in order to protect the identity of reviewers who chose to remain anonymous.

Reviewer 2 ·

Basic reporting

No comment

Experimental design

No comment

Validity of the findings

No comment

Additional comments

Comments to the authors:

Breast cancer is the most commonly occurring cancer in women and the second most common cancer overall. Therefore, studies that indicate new possible therapeutic targets or prognostic biomarker should be encouraged. In the present study, the authors use 6 kinds of immune cells composition analysis, WGCNA expression analysis, and other methods to screen out five macrophage-related hub genes that were associated with prognosis of triple negative breast cancer (TNBC). However, several points are to be considered.

Major comments

1- Although the analysis of the blue module has pointed out the possible existence of other cell types in the tumor, the authors invest only in macrophages. At this point I have some questions:

a. Why did the authors exclude the other cells from the analysis?

b. Under these conditions, the enrichment analysis presented in Figure 1E shows biological processes associated with neutrophil function and other generic processes linked to immune response, but none directly associated with macrophage function. How do the authors explain this fact? Could other cells be involved?

c. Did the authors perform analyzes in other molecular signature databases besides GO?

2. The authors identify five hub genes (CD79A, CXCL13, IGLL5, LHFPL2 and PLEKHF1), which are referred to as macrophage-associated genes. However, it seems not to be very clear the association of these genes with macrophages. According to the Human Protein Atlas (https://www.proteinatlas.org) CD79A is expressed on B lymphocytes, LHFPL2 on dendritic cells and PLEKHF1 on T cells but not on macrophages. Thus, the authors should clarify what supports the statement that these genes are linked to macrophages and not to other cells.

3. Figure 3 is confused; the text order does not match the order of the panels. For example, figure 3D should be figure 3E and figure 3F should be figure 3H. Please check the entire figure 3.

4. Based on the GDSC database(https://www.cancerrxgene.org/) the authors evaluated the responses of two subgroups Cluster A and Cluster B that were previously clustered according to 5 hub gene to anticancer drugs. However, this tool has characterized 1000 human cancer cell lines and screened them with 100s of compounds. Considering that the focus of the study is on genes expressed by immune cells such as macrophages. How can drug testing in tumor cell lines be associated with the genes described in Cluster A and B of the present study? Furthermore, it would be interesting the authors to describe in more detail the methods of how they performed this analysis.


Minor comments

1. Authors could indicate the meaning of acronyms in the abstract. For example, TNBC, BRCA and GDSC are without their description. Additionally, throughout the text this question is repeated. Please review the use of acronyms.

2. Please increase the font size in the figures

3. In figure 2D the authors could indicate the clusters as A and B.

4. Authors could standardize the use of colors to indicate groups. For example, in Figure 2E cluster A is blue and cluster B yellow. Next in figure 2F cluster A is yellow and cluster B is blue. Standardization can help read the article.

Reviewer 3 ·

Basic reporting

The article merits English revision. The authors must mention the limitations of solely using bioinformatics platforms. That information must the included in the discussion section. Regarding the structure of the figures, improving the resolution and quality of all figures is essential. Most of them is of low quality, which makes them unintelligible.

Experimental design

In the experimental design, please add a workflow to depict the study phases and number of samples used in each step.

Validity of the findings

The study provides a relevant research question. However, it is necessary to carry out an external validation of the findings to ensure a robust conclusion. We mean, the use of databases is associated with several limitations. Please, see the comments to the authors.

Additional comments

Su et al. evaluated an individualized gene signature related to macrophages to predict
overall survival in patients with triple negative breast cancer. The authors showed that they found five central prognostic genes associated with macrophages, and a prognostic model was constructed to predict the survival of patients with TNBC. However, the study lacks external validation to improve the study robustness to support the findings of the bioinformatics platforms. Some important points must be addressed to improve the quality of this study before the final recommendation as follows:

Comment:

• Immune infiltration analysis and WGCNA screening of macrophage related genes in TNBC (line 209-210): The authors showed a gene signature in TNBC macrophage infiltrate. What is the immunological profile of this macrophage? M1 or M2? Based on that, an independent external validation of the signature is essential to support the conclusions.

• The following confounding factors were not taken into account. Key points unadressed: 1) the duration of chemotherapy exposure and sample collection. 2) Type of chemotherapy regimen used. 3) Considering the chemotherapy is known to modulate the macrophage phenotype in tumor microenvironment, a group naive of treatment would be essential.

---

## Round 0.2 · Major Revisions

I appreciate the overall improvements on the manuscript, however there are 2 majors points pointed out by reviewer 2 which still remains unsolved. I believe these final corrections could be completed in 2 weeks.

Reviewer 1 ·

Basic reporting

The authors have done a commendable job in increasing the quality of writing this paper. It was much easier to read and follow. Thank you for adding Fig 1 as it helps the reader to understand the paper.

Experimental design

No comment

Validity of the findings

No comment

Additional comments

Su P. et al did extensive analysis and validated their observations with multiple data sets and programs. This study is important in the field of developing personalized medicine against TNBC.

The authors have incorporated all the comments as suggested and it was a delight to read this modified version of the impactful study.

Reviewer 2 ·

Basic reporting

none

Experimental design

none

Validity of the findings

none

Additional comments

Two major points remain unresolved:

1. It must be demonstrated that these 5 hub genes are expressed in macrophages. This can be done by rtPCR on differentiated M0, M1 or M2 macrophages, or use a public database with these cells, or use a single cell analysis of TAMs.

2. Drug testing is done on tumor cells, not macrophages. Are these genes also expressed in tumor cells? As a suggestion, this data could be removed from the article.

---

## Round 0.3 · accepted · Accept

We appreciate your efforts to improve the overall quality of this work. This study represents a relevant contribution on cancer biology microenvironment, an increasing area in which knowledge is crucial to allow new clinical approaches with better responses for patients.

Reviewer 2 ·

Basic reporting

no comment

Experimental design

no comment

Validity of the findings

no comment

Additional comments

The article can be accepted in its current form. However, as a suggestion I believe that use "hub genes associated with immune response" is more adequate.